# Significance of the cGAS-STING Pathway in Health and Disease

**DOI:** 10.3390/ijms241713316

**Published:** 2023-08-28

**Authors:** Jinglin Zhou, Zhan Zhuang, Jiamian Li, Zhihua Feng

**Affiliations:** 1Fujian Key Laboratory of Innate Immune Biology, Biomedical Research Center of South China, College of Life Science, Fujian Normal University Qishan Campus, Fuzhou 350117, China; 2Key Laboratory of College of First Clinical Medicine, College of First Clinical Medicine, Fujian Medical University, Taijiang Campus, Fuzhou 350001, China

**Keywords:** cGAS–STING pathway, anti-pathogen immunity, inflammation, autoimmune disorder, cancer

## Abstract

The cyclic GMP-AMP synthase (cGAS)-stimulator of interferon genes (STING) pathway plays a significant role in health and disease. In this pathway, cGAS, one of the major cytosolic DNA sensors in mammalian cells, regulates innate immunity and the STING-dependent production of pro-inflammatory cytokines, including type-I interferon. Moreover, the cGAS–STING pathway is integral to other cellular processes, such as cell death, cell senescence, and autophagy. Activation of the cGAS–STING pathway by “self” DNA is also attributed to various infectious diseases and autoimmune or inflammatory conditions. In addition, the cGAS–STING pathway activation functions as a link between innate and adaptive immunity, leading to the inhibition or facilitation of tumorigenesis; therefore, research targeting this pathway can provide novel clues for clinical applications to treat infectious, inflammatory, and autoimmune diseases and even cancer. In this review, we focus on the cGAS–STING pathway and its corresponding cellular and molecular mechanisms in health and disease.

## 1. Introduction

The innate immune system, which constitutes part of the defense system of mammals, can recognize the “non-self” genetic material of invading pathogens. In particular, the germline-encoded pattern-recognition receptors (PRRs) that are usually located on the cell surface or in the cellular compartments of the cytosol distinguish the pathogen-associated molecular patterns (PAMPs) of extracellular pathogens [1]. Furthermore, “self” DNA molecules, which are vital genetic material found within specific regions of cells, can occasionally occur as misplaced DNA within the cytosol. This DNA is also recognized by PRRs via damage-associated molecular patterns (DAMPs) [2]. Consequently, the recognition of both PAMPs and DAMPs by PRRs elicits the innate defense mechanism, as well as the production of soluble mediators, such as type-I interferon (IFN) and other cytokines, to impede viral or microbial infection and maintain cellular balance [3]; thus, the production of type-I IFN is fundamentally controlled by PRRs. Among the different PRRs, cyclic GMP-AMP synthase (cGAS) is a class of PRRs involved in recognizing cytosolic DNA. A stimulator of interferon genes (STING), which is localized on the endoplasmic reticulum (ER), was initially discovered as the upstream initiator of type-I IFN [4]; however, STING cannot directly bind to immune-stimulatory DNA (ISD). Instead, bacteria-derived cyclic diguanylate monophosphate (c-dGMP) or cyclic di-adenosine monophosphate (c-dAMP) serve as the upstream ligands of STING [5,6]. Subsequently, cGAMP (cGMP-AMP), which is synthesized by cGAS, acts as the novel second messenger of STING [7].

The cGAS–STING pathway is widely expressed in immune, non-immune, and cancer cells [8]. In addition to triggering the type-I IFN signaling cascade to mediate cellular innate immune responses, the cGAS–STING pathway is also involved in other intrinsic cellular processes, including cell death, autophagy, and cellular senescence [9]. Moreover, the cGAS–STING pathway is regulated by other DNA-sensing pathways and cellular molecules to maintain intracellular homeostasis under normal conditions [9]. 

The primary role of the cGAS–STING pathway is to activate host innate immunity against a DNA-containing pathogen infection [10]; however, abnormal activation of the cGAS–STING pathway can lead to the excessive and continuous production of type-I IFN, resulting in its disproportionate accumulation in tissues and organs [11]. Moreover, increasing evidence has suggested that the abnormal accumulation of type-I IFN is involved in the pathogenesis of autoimmune diseases and mediates inflammation development in other diseases [12,13]. Additionally, activating the cGAS–STING pathway can restrict tumor growth by inducing specific antitumor immunity [11]; therefore, preclinical research targeting the cGAS–STING pathway can pave the way for future translation into clinical trials, ultimately alleviating symptoms, improving the life quality of patients with various autoimmune or inflammatory disorders, and prolonging the life expectancy of patients with cancer. 

In this review, we describe the mechanisms by which the cGAS–STING pathway induces type-I IFN immune responses and the regulation of the cGAS–STING pathway by other DNA-sensing pathways and cellular molecules. We further cover the role of the cGAS–STING pathway in certain cellular processes, such as cell death, cell senescence, and autophagy. Additionally, we highlight the importance of the cGAS–STING pathway in anti-pathogen immunity and summarize the advances in the current research targeting this pathway in various disorders and cancers. Finally, we discuss the existing and potential therapeutic strategies focused on the cGAS–STING pathway and provide recommendations that should be addressed in future studies.

## 2. Non-Canonical Activation of cGAS and cGAMP Synthesis

cGAS is not only localized in the cytosol, but has also been detected in nuclear regions [7,14]. cGAS is a 60 kDa protein with a non-conserved amino-terminal stretch composed of approximately 130–150 residues, along with a highly conserved Mab21 domain belonging to the nucleotidyl transferase (NTase) superfamily and three other major domains, including an unstructured active domain, a two-lobed inactive catalytic domain, and an extended N-terminal domain [15]. 

cGAS functions as a cytosolic DNA sensor; thus, its activation is determined by its recognition of cytosolic double-stranded DNA (dsDNA). Moreover, cGAS can be activated by dsDNA from multiple sources, such as microbial DNA, mitochondrial DNA (mtDNA), nuclear chromatin, extracellular self-DNA, cytosolic chromatin and micronuclei, and dysfunctional telomeres [3]. The dsDNA from microbes, including simplex virus, vaccinia virus, cytomegalovirus, *Chlamydia trachomatis, Mycobacterium tuberculosis,* and *Francisella novicida*, can enter the cytosol and be sensed by cGAS, leading to a downstream signaling cascade that induces the type-I IFN response [10]. Mitochondrial stress can be induced by the deficiency of mitochondrial transcription factor A (TFAM; an mtDNA-binding protein) or the release of mtDNA into the cytosol by herpes viruses, resulting in cytosolic mtDNA that subsequently engages cGAS [16]. Extracellular self-DNA can arise from the deficiencies of three-prime repair exonuclease 1 (TREX1; an exonuclease) or lysosomal DNases and then invade the cytosol of host cells, thereby activating cGAS [17]. Cytosolic micronuclei mainly arise from DNA double-strand breaks (DSBs), mitotic errors, and abnormal DNA replication, which result in acentric chromosome fragments and whole non-segregated chromatins that are generated due to their failure to be included in the daughter nuclei after telophase [18]. Eventually, these chromatins and chromatin fragments recruit their nuclear envelopes (NEs) and form micronuclei in the cytosol. The collapse of the NE of these micronuclei then leads to the recognition of the cytosolic chromatin DNA by cGAS, causing its activation [19]. DNA from dysfunctional telomeres is another potential source of cytosolic DNA. Telomere shortening is a hallmark of cellular senescence, which induces replicative crises, such as mitotic delay and deformation of chromosomes (e.g., chromosomes with two centromeres). This abnormal chromosomal DNA reaches the cytosol and is detected by cGAS [20]. Furthermore, only dsDNA over 40 bp in length, such as those previously mentioned, can theoretically be detected by cGAS; however, a unique structure called “stem-loop” formed during the reverse transcription of human immunodeficiency virus type 1 (HIV-1) has been reported to activate cGAS in a sequence-dependent manner due to the presence of unpaired guanosines flanking short dsDNA (Y-DNA), which are considered the cGAS recognition motif and enhancer of cGAS enzymatic activity [21]. One report indicated that cGAS is predominantly localized in the nuclear region, which is tethered tightly by a salt-resistant interaction in order to maintain the resting state of cGAS and prevent auto-reactivity [14]. In addition, another study illustrated that in the nucleus, the barrier-to-autointegration factor 1(BAF1) competitively binds to self-DNA for prohibition of the formation of DNA–cGAS complexes that are essential for the enzymatic activity of cGAS [22].

After recognizing these DNA stimulants in mammalian cells, each of the two DNA-binding sites on the catalytic domains of cGAS binds with different dsDNA, generating a 2:2 cGAS–dsDNA complex oriented in two different directions that represents the minimal active enzymatic unit [13]. Additionally, in the human cGAS (h-cGAS), the 2:2 cGAS–dsDNA complexes rearrange themselves to create a ladder-like network that enables the successive recruitment of adjacent cGAS to form more 2:2 cGAS–dsDNA and stabilize its structure [23,24]. Next, the catalytic domain of the cGAS–dsDNA complex is rearranged to catalyze its substrates (guanosine triphosphate [GTP] and adenosine triphosphate [ATP]) into cGAMP, which contains two phosphate diester bonds that have a high affinity for STING [25,26,27,28]. Furthermore, the N-terminal of cGAS mainly induces the liquid phase transition of cGAS to form liquid-like droplets with dsDNA that promote cGAMP production by enhancing the concentration of enzymatic activity and reactants [29,30]. This liquid phase transition requires an appropriate concentration of cGAS and dsDNA because cGAS is only activated when dsDNA reaches a certain concentration [26]. 

## 3. Downstream Signaling Reaction of cGAMP

The STING protein comprises four major components, including a short cytosolic N-terminal segment, a four-span transmembrane domain, a connector region, and a cytosolic ligand-binding domain (LBD) with a C-terminal tail (CCT) [13].

cGAMP can be sensed by the ER-located STING [4,31]. After detecting cGAMP, STING undergoes dimerization, followed by the LBD of the STING dimer rotating in the opposite direction to form an ordered β-sheet [32]. Subsequently, the STING dimer forms a “cryo-EM” structure, leading to a structural conformational change that causes self-activation [32,33]. After activating itself, the trafficking STING dimer translocates from the ER to the ER–Golgi intermediate compartment (ERGIC) with the collaboration of the TRAP–translocon complex [34]. Simultaneously, STING recruits TANK-binding kinase 1 (TBK1) for a downstream signaling cascade during this process [13]. Next, cytoplasmic coat protein complex-II (COPII) and translocon-associated protein β (TRAPβ) facilitate STING to reach the Golgi [35]. In the Golgi, the Ser366 site in the CCT of trafficking STING is crosswise phosphorylated by an adjacent TBK1 associated with another STING, while the TBK1 molecules phosphorylate each other [36]. This modification enables the highly efficient activation and palmitoylation of STING, and STING and TBK1 aggregate to form the STING signalosome [37]. Subsequently, STING and TBK1 rearrange to a “scaffold” structure, leading to the phosphorylation of interferon regulatory factor 3 (IRF3). Finally, the phosphorylated IRF3 undergoes dimerization and is trafficked into the nucleus as a transcriptional factor that binds the promoter of type-I IFN, causing its expression [38]. In addition, the STING–TBK1 signalosome can phosphorylate the inhibitors of the transcription factor NF-κB (IκBα), resulting in the polyubiquitination and degradation of IκBα by the ubiquitin–proteasome pathway to release NF-κB into the nucleus [4]. The NF-κB has two main roles in the nucleus. One role involves aiding the expression of type-I IFN, and the other consists of regulating the transcription of inflammatory cytokines, such as interleukin 6 (IL-6) and tumor necrosis factor (TNF) [38,39]. In summary, the non-canonical activation of the cGAS–STING pathway involves the induction of pro-inflammatory cytokine expression, particularly type-I IFN. Activation of the cGAS-STING pathway during innate immune response was showed in the Figure 1.

## 4. Regulation of the cGAS–STING Pathway

The cGAS–STING pathway is positively and negatively regulated by cellular molecules, enzymes, and other DNA-sensing pathways. 

In the negative regulation of the cGAS–STING pathway, the most efficient mechanism is to eliminate cytosolic DNA and thereby prevent cGAS activation. Several cellular nucleases can perform this role. One such nuclease is DNase II, a lysosomal enzyme responsible for DNA digestion in endosomes or autophagosomes to prevent DNA from leaking into the cytosol [17]. Similarly, the main function of TREX1 (also called DNase III) is to eliminate free dsDNA entering the cytosol [40]. Additionally, the RNaseH2 endonuclease complex (RNaseH2A, RNaseH2B, and RNaseH2C) is responsible for separating ribonucleotides embedded in the DNA of RNA–DNA hybrids, which ensures that DNA replication occurs on the rails and restricts the release of abnormally replicated DNA into the cytosol [41]. Another nuclease is SAM domain- and HD domain-containing protein 1 (SAMHD1), which is widely characterized as a dNTPase that acts as an inhibitor to limit the reverse transcription of RNA viruses. Moreover, activated SAMHD1 can increase the enzymatic activity of MRE11 to degrade nascent ssDNA at stalled replication forks, which impedes the accumulation of ssDNA fragments in the cytosol [42]. Consequently, a deficiency in any endonucleases, particularly DNase II, TREX1, the RNaseH2 endonuclease complex, and SAMHD1, may lead to the abnormal accumulation of dsDNA, as well as excessive activation of the cGAS–STING pathway [9].

Apart from the previously mentioned endonucleases, other DNA sensors can also negatively modulate the cGAS–STING pathway. γ-interferon-inducible protein-16 (IFI16) is a typical absent melanoma 2 (AIM2)-like receptor from the PYHIN family, which can sense an invading DNA virus or damaged chromatin DNA in both the nucleus and cytosol. After DNA recognition, IFI16 initiates the STING-dependent type-I IFN signaling cascade with TNF receptor-associated factor 6 (TRAF6) and p53 [43]. Studies have also indicated that IFI16 competitively binds with DNA and adaptor STING, partially suppressing cGAMP production; yhus, IFI16 is essential for the higher-level responses of STING [44,45]. Furthermore, the HIN2 domain of IFI16 inhibits the activation of cGAS, while a portion of cGAS can enter the nucleus and maintain the stability of IFI16 [46,47]. AIM2 is another DNA sensor in the PYHIN family located in the cytosol. After recognizing dsDNA, AIM2 associates with apoptosis-associated speck-like proteins, such as a CARD (ASC) and procaspase-1, to form inflammasomes [48]. The inflammasomes lead to a multi-protein signaling cascade that activates caspase-1 and further facilitates the transformation of pro-inflammatory cytokines, such as IL-1β and IL-18, into active forms, ultimately inducing pyroptosis [49]. Simultaneously, the AIM2-induced inflammasomes interfere with the activation of the cGAS–STING pathway via the cleavage of cGAS and reduce cell viability [50,51]. Furthermore, the activated caspase-1 cleaves gasdermin D and promotes its pore-forming activity, resulting in an intracellular potassium (K^+^) efflux that prevents cytosolic DNA from engaging cGAS [52]. Alternatively, activating the cGAS–STING pathway may dampen other DNA sensor-induced type-I IFN production. Toll-like receptor 9 (TLR9) is an important endosomal PRR composed of a leucine-rich repeat (LRR) domain and a cytosolic c-terminal Toll/IL-1 receptor (TIR) domain [53]. During infection, TLR9 recognizes cytosine–phosphate–guanosine (CpG)-rich sequences formed from bacterial or viral DNA via its LRR domain, which leads to the recruitment of MyD88 to the cytosolic C-terminal TIR domain for the nuclear translocation of IFN transcription factor 7 (IRF7), eventually enhancing type-I IFN production [54]. In plasmacytoid dendritic cells (pDCs), activation of the cGAS–STING pathway triggers the expression of suppressor of cytokine signaling 1 (SOCS1) and SOCS3, which subsequently inhibits the TLR9-mediated type-I IFN signaling cascade [55].

The downstream ligands of cGAS, including cGAMP and STING, can also negatively regulate cGAS activity through modifications such as deubiquitylation, glutamylation, phosphodiesterase-catalyzed hydrolysis, sumoylation, and phosphorylation. In addition, cGAS activity can be modulated by direct protein–protein interactions, while intracellular molecules can also modulate cGAS activity. Table 1 summarizes the cellular molecules that can suppress the activity of cGAS, cGAMP, or STING by modification, direct protein binding, or degradation. 

The positive regulation of the cGAS–STING pathway is mainly achieved by intracellular modulators during the post-translational stages via modifications, direct interactions, or indirect assistance. These modulators are summarized in Table 2.

## 5. cGAS–STING Pathway in Autophagy

Autophagy is an intracellular, self-protective mechanism to maintain energy balance in response to nutrient stress in cells. This process can degrade misfolded or aggregated proteins, damaged organelles, and invading pathogens [82]. Studies have shown that under conditions of pathogen infection, the cGAS–STING pathway can induce autophagy without triggering type-I IFN or NF-κB signaling [83]. Specifically, in mammalian cells, cGAMP-bound STING translocates to the ERGIC independently of the recruitment of TBK1. The STING-containing ERGIC then acts as the membrane source of LC3B (microtubule-associated proteins 1A/1B light chain 3B) lipidation to form autophagosomes that engulf cytosolic DNA, subcellular organelles, or misfolded proteins with the collaboration of WD repeat domain phosphoinositide-interacting protein 2 (WIPI2) and autophagy protein 5 (ATG5) [83]. In contrast, the formation of canonical autophagosomes requires the recruitment of the unc-51-like kinase 1 (ULK1) complex that phosphorylates components of the class III PI3K (PI3KC3) complex I and the negative modulation of the mammalian target of rapamycin (mTOR); therefore, the cGAS–STING pathway-induced autophagy is a novel bypass mechanism to initiate autophagy. Moreover, in certain ancient invertebrate species, such as the anemone *Nematostella vectensis* that emerged 500 million years before human beings, the cGAS–STING pathway-induced autophagy plays an essential role in resisting microbe infection [84]. In the case of mammals, the cGAS–STING pathway-induced autophagy may be mainly aimed at protecting against the invasion of extracellular pathogens, including *M. tuberculosis*, certain Gram-positive bacteria, and the Zika virus; however, its excessive signaling may cause irreversible apoptosis during cellular replicative stress [20,84,85,86,87]. Nevertheless, the cGAS–STING pathway-dependent autophagy is generally conserved and can reduce the over-reaction of the cGAS–STING pathway-induced type-I IFN signaling cascade. 

## 6. cGAS–STING Pathway in Cell Death

Another functional response mediated by the cGAS–STING pathway is triggering cell death. The cGAS–STING pathway mainly initiates three types of cell death pathways (i.e., lysosomal-dependent pyroptosis [LDCP], apoptosis, and necroptosis) to eliminate infected, damaged, or transformed cells and maintain organismal homeostasis.

The hallmark of LDCP is the release of lysosomal hydrolases, such as cathepsins, into the cytosol due to lysosomal membrane permeabilization and the ensuing pyroptosis [88]. In specific cell types, particularly human myeloid cells, STING is sorted into lysosomes after activating the downstream signaling cascade, causing permeabilization of the lysosome membrane and consequent leakage of cathepsins into the cytosol. Next, the NLRP3 inflammasome and caspase-1 are activated, ultimately resulting in a K^+^ efflux and pyroptosis. Pyroptosis has also been suggested to be a secondary inflammatory response in dying cells involving the secretion of cytokines, such as IL-1β and pro-IL-1β [89]. 

Apoptosis is a programmed cell death process that activates neighboring cells or macrophages to eliminate damaged cells [90]. The initiation of apoptosis relies on activating any three signals: extrinsic signals, intrinsic mitochondrial signals, and granzyme-mediated signals [91]. The intrinsic mitochondrial signals, including cellular stresses such as genotoxic stress, activate two proapoptotic proteins, BCL-2-associated X protein (BAX) and BCL-2 homologous killer (BAK), by stimulating only BH3 domain family members (BIM). These activated proapoptotic proteins cooperatively form a pore-like conformation, inducing mitochondrial outer membrane permeabilization, which in turn results in the release of mitochondrial contents, including cytochrome c and mtDNA [92]. Cytochrome c then binds with an adapter apoptotic protease activating factor-1 (APAF1) to form a cytochrome c-APAF1 association. Subsequently, this association recruits and activates caspase-9 to form an apoptosome that triggers apoptosis [93]. Additionally, the activated caspase-9 activates caspase-7 and -3, which results in the inhibition of the cGAS–STING pathway due to the cleavage of cGAS and IRF3 by the activated caspase-7 and -3 [94]. Although the cGAS–STING pathway is inhibited during apoptosis, independent studies have confirmed that certain mechanisms ensure that moderate cGAS–STING-mediated type-I IFN production is triggered by mtDNA in response to pathogen infection [95,96]. Considering the findings above, the function of caspase-9-mediated apoptosis may involve avoiding excessive type-I IFN immune responses. Another study has reported that the infection of *Mycobacterium bovis* in murine macrophages can lead to the release of activated STING into the cytosol due to ER stress [97]. In this context, STING can directly recruit phosphorylated TBK1, which then activates IRF3 that can directly trigger apoptosis in a BAX–BAK-dependent manner [97,98]; thus, the cGAS–STING pathway-induced activation of IRF3 results in apoptosis rather than a type-I IFN signaling cascade under viral infection or irreversible damage. Furthermore, excessive activation of autophagy mediated by cGAS–STING may induce the irreversible apoptosis process termed autophagy-dependent apoptosis [87]. 

Necroptosis is a type of cellular necrosis that serves as a supplementary mechanism for regulating cell death to ensure normal development and homeostasis of multicellular organisms [99]. Excessive production of TNF and type-I IFN has been shown to contribute to necroptosis initiation when apoptosis is moderated by various conditions, including genetic defects, caspase-9 inhibition, and pathogen invasion [100]. Another study illustrated that in bone marrow-derived macrophages (BMDMs), type-I IFN and TNF produced in a cGAS–STING pathway-dependent manner are maintained at a critical threshold level after the recognition of cytosolic DNA. This threshold maintenance facilitates the subsequent activation of mixed lineage kinase domain-like protein (MLKL) or receptor-interacting serine/threonine protein kinase 3 (RIPK3) that causes plasma membrane disruption and eventually enables cells to undergo necroptosis, during which caspase-9 is inhibited [101,102].

## 7. cGAS–STING Pathway in Cellular Senescence

The cell cycle is vital in regulating cell proliferation, growth, and division. Cellular senescence is characterized by the permanent arrest of the cell cycle via telomere erosion or various stress, generally comprising ionizing radiation, oxidative stress, or oncogene signaling [103]. Certain hallmarks can help distinguish senescent cells from normal cells, including an enlarged size, increased granularity, increased senescence-associated (SA)-β-galactosidase activity, an increased level of cyclin-dependent kinase inhibitors, the expression of anti-proliferative molecules (p16 and p21), and DNA damage foci [104]. A report using mouse embryonic fibroblasts (MEFs) demonstrated that when cells undergo senescence by ionizing radiation, changes in the nuclear structure lead to the disintegration of the nuclear lamina (NL), which is responsible for maintaining the NE structure. Furthermore, the chromatin fragments generated by senescence-related DNA damage enter the cytosol, activating cGAS [105]. Interestingly, senescent cells have been indicated to contain higher levels of chromatin fragments than normal cells, and the depletion of cGAS can result in the spontaneous immortalization of senescent cells [105,106]. Senescent cells express various SA secretory phenotype (SASP)-related genes, including those of cytokines, chemokines, and proteases, in an autocrine and paracrine manner, which reinforces cell cycle arrest and mediates chronic inflammation [107]. Studies have further shown that although the expression of SASP-related genes in senescent cells is controlled at multiple levels, the cGAS–STING pathway-mediated expression of several genes is related to the regulation of the SASP-related genes. The cGAS–STING pathway recognizes cytosolic chromatin fragments in senescent cells and mediates the expression of cytokines (IL-6 and TNF-α) and chemokines (Cxcl-10, Cxcl-2, Ccl-3, and Ccl-5) to shape the microenvironment of chronic inflammation in senescent cells [108]. 

Hematopoietic stem cells (HSCs) are usually dormant in the bone marrow niche. A study revealed that HSCs trigger the type-I IFN immune response via the cGAS–STING pathway during the invasion of DNA-containing bacteria in the bone marrow. The stimulation of the type-I IFN immune response, in turn, leads to the exhaustion of HSCs [109]. Moreover, the excessive secretion of type-I IFN in response to DNA damage can accelerate cellular senescence and inhibit cellular function via the activation of p53–p21 signals and increased p16^INK4^ levels in HSCs [110]. Interestingly, another investigation highlighted that long-term HSCs produce a peculiar circular RNA termed “cia-cGAS” under homeostatic conditions in the nucleus, which exhibits a higher binding affinity for cGAS than self-DNA. Furthermore, cia-cGAS was found to suppress cGAS–DNA binding, resulting in the inhibition of cGAS-mediated type-I IFN production and the maintenance of a dormant state [60]. All these findings indicate that the cGAS–STING pathway has a predominant role in facilitating cellular senescence. The cGAS–STING pathway in cell death, autophagy, and senescence was illustrated in Figure 2.

## 8. cGAS–STING Pathway in Anti-Pathogen Immunity

Theoretically, all nucleic acids, such as self-DNA and DNA–RNA hybrids, can be bound by cGAS; therefore, the cGAS–STING pathway is a major innate immune pathway triggered against infections by numerous diverse pathogens, such as DNA viruses, retroviruses, DNA-containing bacteria, and parasites. cGAMP has also been reported as an adjuvant that boosts adaptive immune responses by accelerating antigen-specific T-cell activation and antibody production [10,111]. 

Viral invasion, such as that by the herpes simplex virus (HSV), elicits mtDNA stress, causing detrimental effects on mtDNA stability and eventually leading to mtDNA entry into the cytosol. The resulting cytosolic mtDNA promotes the production of type-I IFN and pro-inflammatory cytokines via the cGAS–STING pathway [16]. Moreover, an infection of the Dengue virus and SARS-CoV-2 also causes the activation of the cGAS–STING pathway through cytosolic mtDNA rather than its genome [112,113]. In the case of retroviruses (including the typical HIV-1 retrovirus), cGAS is recruited to the capsid of HIV-1 through binding of polyglutamine-binding protein 1 (PQBP1), which is the adaptor of cGAS, and then induces a STING-dependent anti-viral immune response. [21,114,115,116]. Nevertheless, viruses have been reported to evolve corresponding proteins to inhibit the cGAS–STING pathway and escape immune surveillance. Table 3 displays these viral proteins from previous literature.

During bacterial invasion, including *M. tuberculosis* infection, cGAS in the macrophages detects *M. tuberculosis* genomic DNA in the cytosol. This recognition not only induces significantly higher levels of type-IFN, but also initiates the ubiquitin-mediated selective autophagy to degrade the bacilli [117]. Furthermore, after a *Burkholderia pseudomallei* or *B. thailandensis* invasion, cGAS can induce host cell fusion, forming multinucleated giant cells (MNGCs) via the bacterial type VI secretion system 5 (T6SS-5). This process occurs without interference from extracellular host defenses, facilitating the diffusion of these *Burkholderia* species among the host cells [118]. Furthermore, mitotic events in the MNGCs are abnormally disrupted, resulting in micronuclei formation. cGAS then recognizes the chromatin fragments in the micronuclei by colocalization and consecutively activates STING-dependent autophagy rather than a type-I IFN signaling cascade, ultimately causing autophagy-dependent apoptosis [119]. Interestingly, during the initial formation of these MNGCs, type-I IFN produced in the cGAS–STING pathway induces the expression of guanylate-binding proteins (GBPs), which reduces MNGC formation by inhibiting bacterial Arp2/3-dependent actin motility [120]; therefore, two cGAS–STING pathway mechanisms of genomic replication inhibition are involved in *B. pseudomallei* or *B. thailandensis* infection. Moreover, during *F. novicida* infection, cGAS together with IFI16 binds cytosolic *Francisella* DNA and induces the highest level of STING-dependent type-I IFN immune response [121]. Conversely, certain bacteria have evolved strategies to limit the activation of the cGAS–STING pathway. These bacteria and their mechanisms have been summarized in Table 4. 

The cGAS–STING pathway is also vital in combating parasites. This role has been particularly highlighted in malaria infections caused by the introduction of parasites of the *Plasmodium* genus into the blood cells after mosquito bites. In the blood cells, the *Plasmodium* parasites create extracellular vesicles (EVs) to preserve their genomic DNA; however, monocytes absorb these EVs and cause the parasitic genomic DNA to leak into the cytosol of the host cells [122]. After detecting the genomic DNA of the *Plasmodium* species, cGAS induces type-I IFN production and facilitates the germinal center (GC)-mediated humoral immunity to eliminate the parasites [123]. Furthermore, when *Toxoplasma gondii* enters normal individuals, these parasites are dealt with by the immune system and rendered dormant in the brain. More specifically, after infection with *T. gondii*, the *T. gondii* genome undergoes rapid replication in the cytosol of cerebral cells, activating cGAS. The *Toxoplasma gondii* dense granule protein 15 (TgGRA15) then facilitates the ubiquitination of STING, further enhancing the production of STING-dependent type-I IFN and chemokines to suppress *T. gondii* growth and establish latent infection [124,125].

**Table 3 ijms-24-13316-t003:** Proteins of various viruses restricting the cGAS–STING pathway.

Virus	Proteins	Targets	Mechanisms	Refs.
Herpesviridae				
HSV-1	UL37 VP22 γ_1_34.5	cGAS cGAS STING	Blocks the synthesis of cGAMP Reduces the activity of cGAS Disrupts STING trafficking	[126] [127] [128]
KSHV	ORF52 vIRF1	cGAS STING	Blocks the synthesis of cGAMP Obstructs STING trafficking	[129] [130]
Coronaviridae				
PEDV	PLP2	STING	Impedes polyubiquitination of STING	[131]
SARS-CoV	PLpro	STING	Disrupts the dimerization of STING	[132]
Flaviviridae				
HCV	NS4B	STING	Blocks the interaction between STING and TBK1	[133]
DENV	NS2B	cGAS	Initiates autophagy to degrade cGAS	[134]
ZIKV	NS1	cGAS	Cleaves cGAS	[135]
Papillomaviridae				
HPV	E7	STING	Decreases the activity of STING	[136]
Adenoviridae				
ADEV	E1A	STING	Decreases the activity of STING	[136]
*Hepadnaviridae*				
HBV	Pol	STING	Impedes polyubiquitination of STING	[137]
Poxviridae				
POXV	Poxins	cGAMP	Hydrolyzes cGAMP	[138]
Retroviridae				
HIV	Vpx	STING	Restrains STING-induced NF-κB signaling	[139]

**Abbreviations:** HSV-1, herpes simplex virus 1; KSHV, Kaposi’s sarcoma-associated herpes virus; PEDV, porcine epidemic diarrhea virus; SARS-CoV, severe acute respiratory syndrome-coronavirus; HCV, hepatitis C virus; DENV, dengue virus; ZIKV, Zika virus; HPV, human papilloma virus; ADEV, adenovirus; HBV, hepatitis B virus; Pol, polymerase; POXV, poxvirus; HIV, human immunodeficiency virus; cGAS–STING, cyclic GMP-AMP synthase-stimulator of interferon genes; cGAMP, cGMP-AMP; TBK1, TANK-binding kinase 1.

**Table 4 ijms-24-13316-t004:** Proteins of different bacteria limiting the cGAS–STING pathway.

Bacteria	Proteins	Targets	Mechanisms	Refs.
GBS	CdnP	CDNs	Hydrolyzes cyclic-di-AMP	[140]
*C. trachomatis*	CpoS	STING	Prevents STING-induced cell death and type-I IFN production	[141]
*Yersinia*	YopJ	STING	Restrains the formation of STING-TBK1 signalosome	[142]
*M. tuberculosis*	CpsA	STING	Inhibits STING-dependent autophagy	[143]

**Abbreviations:** GBS, Group B *Streptococcus*; CdnP, a type of ectonucleotidase; CDNs, cyclic dinucleotides; *C. trachomatis*, *Chlamydia trachomatis*; CpoS, *Chlamydia* promoter of survival inclusion membrane protein; YopJ, *Yersinia* outer protein J; *M. tuberculosis*, *Mycobacterium tuberculosis*; CpsA, a member of the LytR-CpsA-Psr (LCP) protein family; cGAS–STING, cyclic GMP-AMP synthase-stimulator of interferon genes; cGAMP, cGMP-AMP; TBK1, TANK-binding kinase 1.

## 9. cGAS–STING Pathway in Autoimmune Disorders and Inflammation

Although the morbidity factors of various autoimmune disorders remain unclear, the abnormal upregulation of type-I IFN has been linked to a spectrum of autoimmune disorders termed “type-I IFN interferonopathies” [144]; thus, the cGAS–STING pathway, which is primarily responsible for producing type-I IFN, is likely to be involved in these disorders.

One such autoimmune disorder is STING-associated vasculopathy with infantile onset (SAVI), which is characterized by ulcerating acral skin lesions, fever episodes, and lung fibrosis. Patients with SAVI present with IgM (Immunoglobulin M) deposition, variable autoantibody titers, and excessive type-I IFN production [145]. This disease is caused by the gain-of-function mutation in the STING-encoding gene that leads to the spontaneous translocation of STING from the ER to the GR in the absence of stimulation by cGAMP, resulting in persistent IRF3 phosphorylation and type-I IFN expression [145]. Another autoimmune disorder is COPA syndrome, which is an early-onset autosomal disease distinguished by arthritis and interstitial lung disease, along with pulmonary hemorrhage as a striking feature. Patients with COPA syndrome have increased levels of T helper cell 17 (Th17 cell) and abnormal expression of several cytokines, such as IL-1β, IL-6, and type-I IFN [146]. This syndrome is attributed to heterozygous mutations in a subunit of coat protein complex I (COPI), resulting in impaired binding and sorting of the proteins targeted for ER retrieval [147]. Consistent with this process, STING is abnormally accumulated in the ER, inducing the intensive type-I IFN signaling cascade [148]. Additionally, familial chilblain lupus (FCL) is a form of cutaneous lupus erythematosus with onset in childhood. Patients with FCL experience cold-induced bluish-red skin lesions in peripheral locations and manifest elevated levels of mucin formation and immunoglobulin deposits [149]. FCL is caused by the heterozygous gain-of-function mutation in STING, which enables STING to undergo dimerization without cGAMP interaction and constitutively activates the type-I IFN signature [150]. In addition to the previously mentioned autoimmune disorders, systemic lupus erythematosus (SLE) is a chronically autoimmune disorder with systemic manifestations, including fever, fatigue, multiple organ failure, and perturbations of the hematopoietic system [151]. Patients with SLE demonstrate deposition of the immune complex formed by anti-nuclear autoantibodies in tissues and abnormal T-cell activation; however, limited evidence has suggested that the cGAS–STING pathway has a central role in SLE progression, despite SLE being a multifactorial disease. Nevertheless, some studies have reported that the loss-of-function mutations in TREX1, RNaseH2C, and DNase I can lead to the aberrant accumulation of cytosolic DNA, which amplifies the type-I IFN signaling cascade in the cGAS–STING pathway [40,152,153]. Moreover, other studies of the serum of patients with SLE have indicated that a defect in eliminating apoptotic cells by macrophages results in the formation of apoptosis-derived membrane vesicles (AdMVs), which contain a significant amount of dsDNA. These AdMVs then induce increased production of type-I IFN to trigger an immune response that causes tissue damage in multiple organs and the regeneration of AdMVs, triggering a positive-loop of type-I IFN production [154]. Aicardi–Goutières syndrome (AGS) is another autoimmune disorder that presents as a progressive brain disease with onset in infancy. AGS is manifested by leukoencephalopathy with basal ganglia calcifications and progressive cerebral atrophy, as well as symptoms overlapping with SLE [41]. Furthermore, patients with AGS exhibit chronic cerebrospinal fluid lymphocytosis and immune complex deposition in tissues, similar to SLE [41]. AGS is caused by autosomal recessive heterogeneous mutations in any nine distinct genes, including LSM11, RNU7-1, TREX1, the RNaseH2 endonuclease complex (RNaseH2A, RNaseH2B, and RNaseH2C), SAMHD1, ADAR, and IFIH1, which are responsible for clearing diverse cytosolic genomic fragments [155,156]. Another study reported that approximately 25% of patients with AGS have mutations in the RNaseH2 endonuclease complex [157]. Furthermore, research has indicated that the loss-of-function mutations in the RNaseH2 catalytic core may adversely affect genome stability, resulting in the presence of cytosolic dsDNA and ensuing excessive cGAS–STING pathway-dependent expression of type-I IFN [158,159]. Finally, the autoimmune disorder rheumatoid arthritis (RA) is a chronic systemic inflammation typically associated with irreversible joint damage, disability, and cardiovascular comorbidities [160]. The etiology of RA is correlated with increased expression of several factors, including pro-inflammatory cytokines, chemokines, and matrix metalloproteinase (MMP), accompanied by the aberrant activation of T cells [161]. Moreover, the accumulation of cytosolic DNA plays a central role in the negative regulation of the inflammatory response in fibroblast-like synoviocytes (FLS) of patients with RA. This is because cytosolic DNA accumulation leads to the continuous activation of the cGAS–STING-dependent type-I IFN signaling cascade and the expression of cytokines and chemokines [161]. 

The inflammatory responses induced by the cGAS–STING pathway also contribute to the development of inflammation in other diseases. The first disease is amyotrophic lateral sclerosis (ALS), which has been suggested to be mainly caused by a mutation in the nuclear DNA/RNA binding protein called TAR DNA-binding protein 43 (TDP-43) [162]. Particularly, the missense mutation in a low-complexity glycine-rich region in TDP-43 enables TDP-43 to accumulate in the cytosol, and it is then absorbed into the mitochondria by mitochondrial import inner membrane translocase 22 (TIM22), causing the leakage of mtDNA into the cytosol via the permeability transition pore. The cytosolic mtDNA is then sensed by cGAS, triggering the production of type-I IFN and NF-κB to initiate hyperinflammatory responses [163]. The second disorder is Huntington disease (HD), which is caused by a mutation of the N-terminal polyglutamine in the huntingtin protein (mHTT) that induces damage to the brain’s striatum [164]. Research has suggested that the striatum injury is associated with DNA damage and upregulation of cGAS in HD cells, triggering type-I IFN inflammatory responses and STING-dependent autophagy [165]. The third condition is myocardial infarction (MI), wherein massive synchronous cell death in the heart and strong cardiac inflammatory response in related tissues are observed during the post-MI period [166]. Additionally, synchronously dying cells can be sensed by cardiac macrophages. This recognition by the cardiac macrophages induces the robust activation of the type-I IFN axis via cGAS–STING, which then maintains an inflammatory microenvironment by inhibiting the transformation of cardiac macrophages from inflammatory cells to a reparative phenotype [167]. The fourth disease is acute pancreatitis (AP), in which many pancreatic acinar cells die, leading to severe inflammation [168]. Furthermore, an independent investigation revealed that DNA released into circulation from necrotic pancreatic cells is recognized and internalized by leukocytes, triggering the cGAS–STING pathway to promote inflammation [169]. The fifth disorder showing inflammation development associated with the cGAS–STING pathway is silicosis. In silicosis, silica microparticles invade the lungs and adhere to lung parenchyma during inhalation, resulting in chronic progressive fibrotic inflammation in the lungs and other complications [170]. Moreover, the etiopathology of lung inflammation in patients with silicosis is associated with a significant increase in the release of chromatin fragments and mtDNA from dying cells into the bronchoalveolar lavage fluid (BALF) after silica exposure, inducing excess production of type-I IFN and CXCL10 via cGAS–STING activation [171]. The sixth ailment associated with the cGAS–STING pathway is nonalcoholic steatohepatitis (NASH). Lipotoxicity in NASH leads to mitochondrial stress and the release of mtDNA into the cytosol of Kupffer cells, which activates cGAS–STING to trigger adipose tissue inflammation in liver that further causes liver obesity, insulin resistance (IR), and glucose intolerance [172,173,174]. Finally, liver ischemia-reperfusion injury (IRI) is the last disorder to be linked with the cGAS–STING pathway. IRI has been reported to induce the generation of reactive oxygen species (ROS) in Kupffer cells, resulting in oxidative mitochondrial damage and the release of mtDNA into the cytosol, triggering the stimulation of NLRP3 inflammasome-driven cell death via cGAS–STING activation [175]. Additionally, another study indicated that the cGAS-STING pathway plays an important role in aging-related inflammation and neurodegeneration [176]. In aged microglia, an increased abundance of mtDNA in cytosol activates chronic inflammatory responses through the cGAS–STING pathway, which contributes to the impaired neuronal function, ensuing damaged brain homeostasis [176]. The different roles of the cGAS–STING pathway in host defense and various diseases were illustrated in Figure 3.

## 10. cGAS–STING Pathway in Cancer

The cGAS–STING pathway plays an indispensable role in radiotherapy, chemotherapy, and anti-tumor immune checkpoint therapy [177,178,179]. The paradigm of cancer is that irreparable damage occurs in normal tissue, which is strongly linked with persistent inflammation [180]. Furthermore, the cGAS–STING pathway has been proposed to not only contribute to the restriction of tumorigenesis, but also facilitate metastasis under certain conditions [181]. 

Moreover, cancer cells are packed with ectopic cytosolic dsDNA, and its accumulation may cause the formation and rupture of micronuclei [182]. In addition, oxidative stress and mitochondrial dysfunction induce the release of mtDNA into the cytosol, thus acting as another source of cytosolic DNA in malignant cells [183]. This cytosolic dsDNA is then recognized by cGAS, and tumor-suppressing biological activities are initiated. A study on early neoplastic progression demonstrated that after cGAS binds to cytosolic dsDNA in cancer cells, the cGAS–STING pathway mediates the secretion of type-I IFN to elicit a robust immune response and recruits immune cells to inhibit tumorigenesis in a cancer cell-autonomous manner [184]. Moreover, tumor DNA and cGAMP can be delivered into tumor-infiltrating dendritic cells (DCs) by tumor-derived exosomes (TEX) and the gap junction, respectively, which results in the production of type-I IFN that facilitates cross-priming with anti-tumor CD8^+^ T cells for the elimination of tumorigenic cells [185,186,187]. Additionally, cGAMP formed in melanoma cells can be transported to proximal non-tumor bystander cells via the gap junction, which results in the activation of STING in these cells and the recruitment of natural killer (NK) cells into the tumor tissue, thereby enhancing tumor regression [188,189]. Activation of the cGAS–STING pathway also induces the expression of anti-proliferative molecules that facilitate senescence in cancer cells [177]. Furthermore, autophagy induced by cGAS–STING activation can halt the transformation of normal cells into cancerous cells by initiating autophagy-dependent cell death in response to abnormal mitotic processes in normal cells [20].

The cGAS–STING pathway can also contribute to metastasis in a non-cell-autonomous manner. A study of patients with metastatic breast or lung cancers found that cGAMP in cancer cells is transferred to astrocytes via the gap junctions, which then activates astrocytic STING to stimulate inflammatory cytokine production. This production of inflammatory cytokines further causes the paracrine activation of the STAT1 and NF-κB pathways in brain metastatic cells, ultimately facilitating the development of metastatic brain cancer [190]. Another research study revealed that the chronic stimulation of the STING pathway contributes to the growth of a 7, 12-dimethylbenz(a)anthracene (DMBA)-induced skin tumor, indicating that STING activation must be maintained appropriately to avoid inflammation-driven tumorigenesis [191]. Moreover, etoposide, camptothecin, and H_2_O_2_ treatment have been shown to cause DNA damage and recruit cGAS to translocate into the nucleus in normal cells. The nuclear cGAS then attenuates the DNA damage response (DDR) mediated by homologous recombination, thereby facilitating cancer development in these cells [192].

Conversely, tumor cells can also escape the immune surveillance mediated by the cGAS–STING pathway. In specific cancer cell lines, the promoters of cGAS and STING are prone to undergo loss-of-function mutation or epigenetic silencing, which leads to the suppression of the cGAS–STING pathway [193]. In addition, the increased lactic acid levels in the lung tumor microenvironment have been revealed to inhibit STING-dependent type-I IFN production in DCs, hampering the capacity of tumor-conditioned DCs to present tumor-associated antigens [194]. Additionally, in breast cancer, activating the human epidermal growth factor receptor 2-RAC-alpha serine/threonine protein kinase (HER2-AKT1) axis inhibits cGAS and TBK1 enzymatic activity [56,195]. Some cancer cell lines also cause proliferation by a unique mechanism that utilizes extrachromosomal telomere repeat (ECTR) DNA to extend telomeres via the alternative lengthening of the telomeres (ALT) pathway [196], which additionally suppresses STING expression [197]. The cGAS-STING pathway was showed in Figure 4.

## 11. Therapeutic Strategies Targeting the cGAS–STING Pathway

As discussed in this review, the cGAS–STING pathway is involved in the pathogenesis of various diseases; therefore, drugs targeting cGAS–STING may provide therapeutic options. For example, inhibitors that negatively modulate the cGAS–STING pathway may be used to reduce the development of autoimmune disorders and local inflammation. In contrast, agonists that enhance the cGAS–STING pathway can improve immune responses and restrict the invasion of extracellular pathogens. In line with this notion, STING agonists can be applied as vaccine adjuvants against infectious diseases [10]. In the case of cancer therapy, cancer immunotherapy targeting the cGAS–STING pathway depends on multiple factors, such as the immune state, the magnitude of the cGAS–STING pathway, different cancer cell types, and tumor stages. Furthermore, the appropriate activation of the cGAS–STING pathway in immune cells provides beneficial effects that can restrain tumor growth, whereas the persistent activation of this pathway may contribute to the formation of carcinogen-induced tumors and arrest the development of T cell-driven adaptive immunity [198]; thus, examining the context of the cancer conditions is crucial in determining the applicability of the cGAS–STING pathway agonists in cancer immunotherapy. Nevertheless, the threshold of cGAS or STING activities must be maintained at normal levels to ensure effective anti-tumor responses and avoid facilitating the development of malignancy and side effects; therefore, developing STING agonists as adjuvants of cancer vaccines may be beneficial [199].

Inhibitors targeting the cGAS–STING pathway can be divided into two categories: cGAS inhibitors and STING inhibitors. The cGAS inhibitors include catalytic site inhibitors that block the activate site of cGAS and DNA-binding inhibitors that competitively bind to dsDNA to inhibit the interaction between DNA and cGAS. The STING inhibitors target the cyclic dinucleotide (CDN)-binding site and act as competitive antagonists of STING activators and palmitoylation inhibitors, which bind with the palmitoylation sites on STING to reduce the recruitment of downstream signal factors and the transformation of STING [13]. Table 5 summarizes these inhibitors and their biological effects. In terms of the cGAS–STING pathway agonists, studies have concentrated on the STING agonists due to their anti-tumor effect. These agonists include CDN, non-CDN, and indirect agonists [200]. Table 6 displays the typical STING agonists and their preclinical effects.

## 12. Conclusions and Future Prospects

Since the discovery of the cGAS–STING pathway, numerous advances have been made in identifying its components the mechanisms by which DNA activates this pathway to induce the secretion of type-I IFN; however, further research is required to address the inconsistent findings regarding the binding of cGAS to ligands from different sources and the underlying subcellular compartmentalization of cGAS. Furthermore, large gaps still exist in understanding the modulation patterns of the cGAS–STING pathway because these alterations involve various modifications by numerous intracellular molecules and signaling networks during the post-translational stages; therefore, future investigations targeting the modulation patterns of the cGAS–STING pathway are required. Moreover, additional studies are required to explore the physiological roles of cGAS and its potential implementation in other intracellular processes involving the induction of type-I IFN.

In summary, our review indicates that cGAS has been well-established as a major sensor that activates the immune response against pathogens. In addition, research on the cGAS–STING pathway can help its application as a novel treatment strategy for cancer, considering its role in inducing cell-intrinsic processes, such as autophagy, cellular senescence, and cell death, or in enhancing innate immunity for the activation of adaptive immunity. In contrast, aberrant or excessive activation of the cGAS–STING pathway can cause primary pathogenesis and the manifestation of several autoimmune diseases; thus, identifying compounds, delivery pathways, and treatment regimens targeting the suppression of the cGAS–STING pathway can provide novel approaches to alleviate the symptoms of autoimmune disorders or inflammation. Conversely, immunopharmacological research on developing cGAS–STING pathway agonists can help formulate therapies for infectious diseases and cancer, such as anti-pathogen or anti-cancer drugs and adjuvants; however, these drugs should be optimized to augment the desirable effect and should not induce any unwanted effects. A reasonable approach may involve controlling the pharmacokinetic parameters appropriately to avoid chronic inflammation triggered by cGAS–STING-induced cytokine storms and to ensure that the drugs reach the target area without side effects. Additionally, considering that discrepancies may arise during drug development due to the differences in the component structures of the cGAS–STING pathway between humans and other species, the preclinical drugs should be carefully examined to determine their suitability in clinical trials. Nevertheless, we anticipate that the development of drugs targeting the cGAS–STING pathway and their translation into clinical applications in the near future will help further ameliorate the condition of patients with cancer, inflammation, and autoimmune diseases.

## Figures and Tables

**Figure 1 ijms-24-13316-f001:**
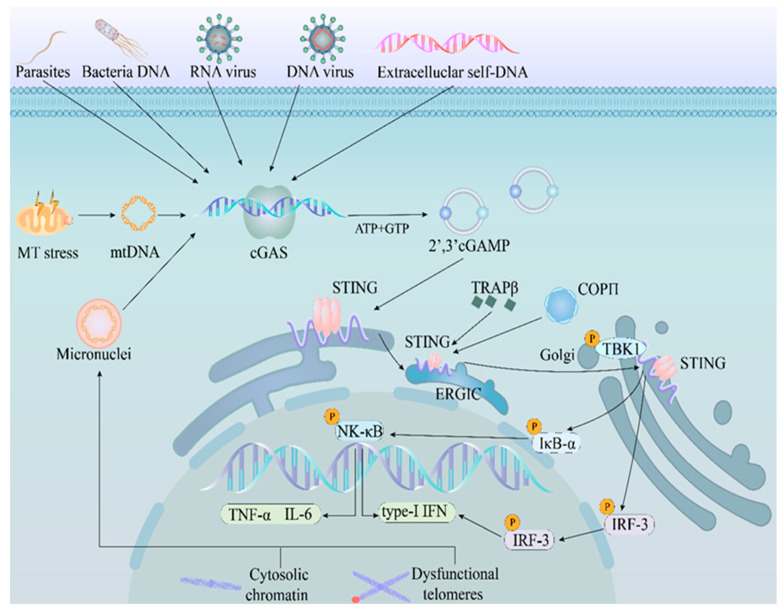
Activation of the cGAS–STING pathway during innate immune response. The DNA from intracellular and extracellular sources is sensed by cGAS. After DNA recognition, cGAS catalyzes its substrates, GTP and ATP, into cGAMP. cGAMP is then sensed by STING located at the ER. Upon cGAMP binding, STING translocates from the ER to the ERGIC and recruits TBK1, followed by which STING and TBK1 are trafficked to the Golgi with the assistance of COPII and TRAPβ. Finally, STING and TBK1 interact with IRF3 and IκBα, culminating in type-I IFN and cytokine production. **Abbreviations:** mtDNA, mitochondrial DNA; cGAS, cyclic GMP-AMP synthase; 2′3′-cGAMP, 2′3′-cGMP-AMP; STING, stimulator of interferon genes; ER, endoplasmic reticulum; ERGIC, ER–Golgi intermediate compartment; TRAPβ, translocon-associated protein β; COPII, cytoplasmic coat protein complex-II; TBK1, TANK-binding kinase 1; IRF-3, interferon regulatory factor-3; IκBα: inhibitors of transcription factor NF-κB; type-I IFN, type-I interferon; IL-6, interleukin 6; TNF-α, tumor necrosis factor-α; GTP, guanosine triphosphate; ATP, adenosine triphosphate.

**Figure 2 ijms-24-13316-f002:**
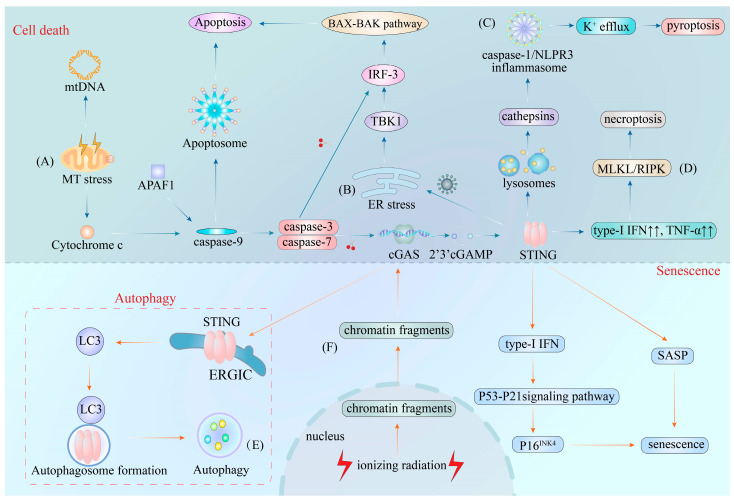
The cGAS–STING pathway in cell death, autophagy, and senescence. (**A**) The cGAS–STING pathway in MT stress-derived apoptosis. Permeabilization of the mitochondrial outer membrane induces the release of mtDNA and cytochrome c into the cytosol. The mtDNA then triggers the activation of the cGAS–STING pathway, leading to the production of type-I IFN and the association of cytochrome c with APAF1 to form the caspase-9-induced apoptosome. This structure further drives apoptosis and promotes caspase-3 and -7 production, which in turn causes the cleavage of cGAS and IRF-3. (**B**) The cGAS–STING pathway in virus-derived apoptosis. During infection with extracellular pathogens, the pathogen-related ER stress results in the direct recruitment of phosphorylated TBK1 by STING to trigger apoptosis via a BAX–BAK-dependent pathway. (**C**) The cGAS–STING pathway in pyroptosis. In specific cell types, STING is sorted into lysosomes after the activation of the cGAS–STING pathway, which leads to the permeabilization of the lysosome membrane and the release of cathepsins into the cytosol. Next, the cytosolic cathepsins trigger the activation of caspase-1 and NLRP3 inflammasome to drive K^+^ efflux-dependent pyroptosis. (**D**) The cGAS–STING pathway in necroptosis. The activation of the cGAS–STING pathway by mtDNA leads to the overexpression of type-I IFN and TNF-α. This overexpression further stimulates the activation of MLKL and RIPK to execute necroptosis. (**E**) The cGAS–STING pathway in autophagy. To prevent the overexpression of type-I IFN, STING traffics to the ERGIC after binding to cGAMP, independently of the recruitment of TBK1. This STING-containing ERGIC then acts as a source of LC3B, which is essential to autophagosome formation to induce autophagy. (**F**) The cGAS–STING pathway in senescence. When cells undergo ionizing radiation, the NL cannot maintain the NE structure, leading to the leakage of chromatin fragments (created by senescence-associated DNA damage) from the nucleus to the cytosol to initiate the overproduction of type-I IFN and SASP via the cGAS–STING pathway. Consequently, the overproduction of type-I IFN activates the p53-p21 signaling pathway to increase p16^INK4^ levels, a hallmark of cellular senescence. **Abbreviations:** MT stress, mitochondrial stress; mt DNA, mitochondrial DNA; APAF1, adapter apoptotic protease activating factor-1; BAX, BCL-2 associated X protein; BAK, BCL-2 homologous killer; K^+^ efflux, potassium efflux; type-I IFN, type-I interferon; TNF-α, tumor necrosis factor-α; MLKL, mixed lineage kinase domain-like protein; RIPK3, receptor-interacting serine/threonine protein kinase 3; LC3B, microtubule-associated proteins 1A/1B light chain 3B; SASP, senescence-associated secretory phenotype; cGAS–STING, cyclic GMP-AMP synthase-stimulator of interferon genes; IRF-3, interferon regulatory factor-3; ER, endoplasmic reticulum; TBK1, TANK-binding kinase 1; ERGIC, ER–Golgi intermediate compartment; NL, nuclear lamina; NE, nuclear envelope.

**Figure 3 ijms-24-13316-f003:**
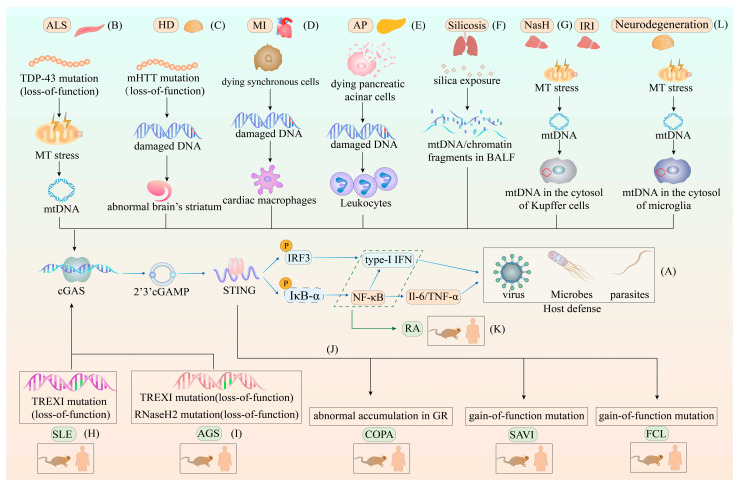
The different roles of the cGAS–STING pathway in host defense and various diseases. (**A**) The role of the cGAS–STING pathway in host defense. The genome of invading pathogens, such as viruses, bacteria and parasites, in the cytosol can be sensed by cGAS, triggering the cGAS–STING pathway-induced production of type-I IFN and other cytokines to eliminate the genomes of invading pathogens. (**B**) The role of the cGAS–STING pathway in ALS. The missense mutation in the TDP-43 protein enables its absorption into the mitochondria and causes mtDNA leakage into the cytosol, prompting overproduction of type-I IFN and NF-κB to initiate hyperinflammatory responses. (**C**) The role of the cGAS–STING pathway in HD. The mutation of the N-terminal polyglutamine in the huntingtin protein (mHTT) leads to the abnormal accumulation of damaged DNA in the brain’s striatum, initiating the STING-dependent type-I IFN production and autophagy. (**D**) The role of the cGAS–STING pathway in MI. Synchronously dying cells are sensed by cardiac macrophages, and damaged DNA from these cells is internalized by cardiac macrophages. Furthermore, this internalization process leads to the inhibition of the transformation of cardiac macrophages from inflammatory cells to a reparative phenotype via the activation of the cGAS–STING pathway. (**E**) The role of the cGAS–STING pathway in AP. Damaged DNA released by the dying pancreatic acinar cells is internalized by leukocytes, resulting in severe inflammation due to the cGAS–STING pathway-induced overproduction of type-I IFN and other cytokines. (**F**) The role of the cGAS–STING pathway in silicosis. Silica microparticles invade the lungs, causing an increased release of chromatin fragments and mtDNA from dying cells into the BALF. These chromatin fragments and mtDNA then engage the cGAS–STING pathway to increase type-I IFN and other cytokine levels, causing chronic progressive fibrotic inflammation in the lungs. (**G**) The role of the cGAS–STING pathway in NASH and liver IRI. Lipotoxicity in NASH and ROS in IRI both lead to mitochondrial stress and the release of mtDNA into the cytosol of Kupffer cells. Furthermore, these changes result in adipose tissue inflammation via the cGAS–STING pathway and cGAS–STING–NLRP3 inflammasome-driven cell death in NASH and IRI, respectively. (**H**) The role of the cGAS–STING pathway in SLE. The loss-of-function mutation in TREX1 is suggested as the main cause of SLE. TREXI mutations lead to the abnormal accumulation of cytosolic DNA, which continuously activates the type-I IFN signaling cascade and amplifies inflammation via cGAS–STING. (**I**) The role of the cGAS–STING pathway in AGS. The loss-of-function mutation in TREX1 and the RNaseH2 complex are considered the major components of AGS. In this context, mutations in TREX1 and the RNaseH2 complex lead to the failure to eliminate genomic fragments in the cytosol and maintain genome stability, causing increased cGAS–STING pathway-dependent expression of type-I IFN. (**J**) The role of the cGAS–STING pathway in COPA, SAVI, and FCL. COPA is caused by the abnormal accumulation of STING in the GR. SAVI is attributed to the gain-of-function mutation in the STING-encoding gene, which leads to the spontaneous translocation of STING from the ER to the GR. This abnormal translocation activates intensive type-I IFN signals independently of cGAMP stimulation. Finally, FCL is caused by the heterozygous gain-of-function mutation in STING, resulting in the spontaneous dimerization of STING and constitutive activation of the type-I IFN signature. (**K**) The role of the cGAS–STING pathway in RA. The abnormal accumulation of cytosolic DNA leads to the expression of cytokines and chemokines via cGAS–STING activation, contributing to RA development. (**L**) The role of the cGAS–STING pathway in neurodegeneration. In microglia, mtDNA in the cytosol of microglia contributes to the chronic inflammatory response, which leads to neurodegeneration. **Abbreviations:** ALS, amyotrophic lateral sclerosis; MT stress, mitochondrial stress; mt DNA, mitochondrial DNA; TDP-43 protein, TAR DNA-binding protein 43; HD, Huntington disease; mHTT, mutation of the N-terminal polyglutamine in huntingtin protein; MI, myocardial infarction; AP, acute pancreatitis; BALF, bronchoalveolar lavage fluid; NASH, nonalcoholic steatohepatitis; IRI, ischemia-reperfusion injury; SLE, systemic lupus erythematosus; AGS, Aicardi–Goutières syndrome; COPA, COPA syndrome; SAVI, STING-associated vasculopathy with infantile-onset; FCL, familial chilblain lupus; RA, rheumatoid arthritis; cGAS–STING, cyclic GMP-AMP synthase-stimulator of interferon genes; type-I IFN, type-I interferon.

**Figure 4 ijms-24-13316-f004:**
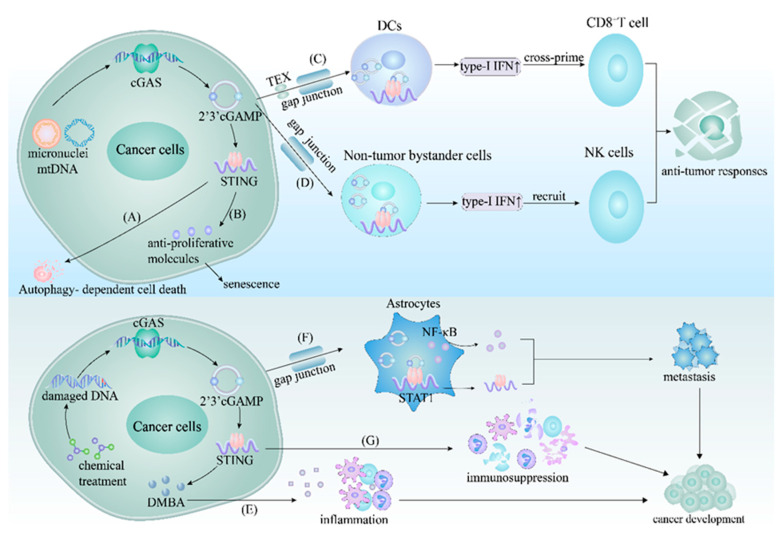
The cGAS–STING pathway in cancer. (**A**) In certain cancers, the cGAS–STING pathway induces autophagy-dependent cell death to limit the transformation of normal cells into cancerous cells. (**B**) The cGAS–STING pathway inhibits the growth of certain cancer cells by inducing the secretion of anti-proliferative molecules that promote senescence. (**C**) In specific cancers, 2′3′-cGAMP from the cancer cells enters the cytosol of DCs through the gap junction or TEX, activating STING to produce type-I IFN. The type-I IFN from the DCs then cross-primes CD8^+^ T cells to eliminate tumorigenic cells. (**D**) Within some cancers, 2′3′-cGAMP from the cancer cells is transferred into non-tumor bystander cells via the gap junction and activates STING, resulting in the recruitment of NK cells to restrain tumor growth. (**E**) In the case of certain cancer cells, chemical treatments cause the introduction of damaged DNA into the cytosol. This damaged DNA further engages the cGAS–STING pathway, inducing the secretion of DMBA to facilitate inflammation and maintain the tumor microenvironment. (**F**) STING activation also occurs after 2′3′-cGAMP from specific cancer cells are trafficked to astrocytes via the gap junction. The activated STING in the astrocytes promotes STAT1 and NF-κB production, further facilitating the development of metastatic cancer cells. (**G**) The loss of function of cGAS or STING leads to immunosuppression in cancer cells due to the hindered production of type-I IFN and other cytokines. **Abbreviations:** TEX, tumor-derived exosomes; DCs, infiltrating dendritic cells; NK cells, natural killer cells; DMBA, 7, 12-dimethylbenz(a)anthracene; cGAS–STING, cyclic GMP-AMP synthase-stimulator of interferon genes; 2′3′-cGAMP, 2′3′-cGMP-AMP; type-I IFN, type-I interferon.

**Table 1 ijms-24-13316-t001:** Intracellular modulators that suppress the activity of cGAS, cGAMP, and STING.

Target	Modulators	Modification	Mechanisms	Refs.
cGAS	AKT	Phosphorylation	Phosphorylates the active domain of cGAS	[56]
	BTK		Reduces the binding efficiency of DNA	[57]
	Caspase-1	Cleavage (degradation)	Dampens the production of type-I IFN	[51]
	TRIM38	Sumoylation	Inhibits the activation of cGAS without microbial invasion	[58]
	TTLL6	Polyglutamylation	Impedes the cGAS–DNA synthesis activity	[59]
	TTLL4	Monoglutamylation	Reduces the enzymatic activity of cGAS	[59]
	Cia-cGAS	Direct binding	Prevents the binding of cGAS to DNA	[60]
	OASL	Direct binding	Decreases the synthase activity of cGAS	[61]
cGAMP	ENP11	Phosphodiesterase	Hydrolyzes cGAMP	[62]
STING	USP13	Deubiquitylation	Prevents the recruitment of TBK1 by STING	[63]
	NLRC3	Direct binding	Impairs the interaction between STING and TBK1	[64]
	NLRX1	Direct binding	Impairs the interaction between STING and TBK1	[65]
	ULK1	Phosphorylation	Thwarts sustained type I IFN production by STING	[66]
	RNF5	Ubiquitylation	Degrades STING at mitochondria	[67]
	AP-1	Directly binding	Terminates the sorting of STING in Golgi	[68]
	2-BP	Directly binding	Suppresses palmitoylation of STING in Golgi	[69]

**Abbreviations:** AKT, RACα serine/threonine-protein kinase (also known as PKB); BTK, Bruton’s tyrosine kinase; ENPP1, ectonucleotide pyrophosphatase/phosphodiesterase family member 1; TRIM38, tripartite motif 38; TTLL6, tubulin-tyrosine ligase-like member 6; TTLL4, tubulin-tyrosine ligase-like member 4; USP13, ubiquitin specific peptidase 13; cia-cGAS, circular RNA with higher cGAS affinity than dsDNA; OASL, 20–50 oligoadenylate synthase; NLRC3, NOD-like receptor family 3 containing CARD domain; NLRX1, NLR family member X1; ULK1, serine/threonine UNC-51-like kinase (ULK1/ATG1); RNF5, RING finger domain; AP-1, adaptor protein complex 1; 2-BP, 2-bromopalmitate; cGAS, cyclic GMP-AMP synthase; cGAMP, cGMP-AMP; STING, stimulator of interferon genes.

**Table 2 ijms-24-13316-t002:** Intracellular modulators that enhance the activity of cGAS, cGAMP, and STING.

Target	Modulators	Modification	Mechanisms	Refs
cGAS	SENP7	Desumoylation	Cleaves sumoylated cGAS to increase its activity	[70]
	HDAC3	Deacetylation	Enhances the activity of cGAS	[71]
	RNF185	E3 ubiquitylation	Promotes the catalytic activity of cGAS	[72]
	Mn^2+^ ion	Direct binding	Augments cGAMP–STING binding affinity	[73]
	G3BP1	Direct binding	Required for the high-level activation of cGAS	[74]
	R848	Indirect assistance	Enables the recognition of HIV-1 infection by cGAS	[75,76]
	PAM3	Indirect assistance	Enables the recognition of HIV-1 infection by cGAS	[75,76]
	TRIM41	Monoubiquitylation E3 ubiquitylation	Promotes the activation of cGAS against viruses infection Protects cGAS degradation from autophagy	[77,78]
cGAMP	Zn^2+^ ion	Direct binding	Accelerates cGAS–DNA synthesis for liquid phase condensation	[29]
STING	AMFR	E3 ubiquitylation	Facilitates recruitment of TBK1 by STING and its translocation	[79]
	MUL1	Polyubiquitylation	Promotes STING trafficking	[80]
	TMEM203	Direct binding	Cooperates with STING to activate TBK1 and IRF3	[81]

**Abbreviations:** SENP7, sentrin/SUMO-specific protease 7; HDAC3, histone deacetylase 3; RNF185, RING finger protein 185; AMFR, autocrine motility factor receptor; MUL1, mitochondrial E3 ubiquitin protein ligase 1; Zn^2+^ ion, zinc ion; Mn^2+^ ion, manganese ion; G3BP1, GTPase-activating protein SH3 domain-binding protein 1; TMEM203, transmembrane protein 203; R848, TLR7/8 agonist; PAM3, TLR2 agonist,; TRIM41, tripartite motif protein 41; cGAS, cyclic GMP-AMP synthase; STING, stimulator of interferon genes.

**Table 5 ijms-24-13316-t005:** Inhibitors of cGAS or STING.

Compound	Characteristics	Biological Effects	Refs
cGAS inhibitors			
Catalytic site inhibitors			
PF-06928215	Binds to the active site of cGAS occupied by ATP	Attenuates type-I IFN signaling in AGS mouse models	[201,202]
RU.521	Competitively occupies the catalytic site of cGAS	Reduces expression levels of *Ifnb1* in BMDMs	[203]
G150	Binds to the active site of cGAS	The IC_50_ is 0.62 μM in primary H-macrophages	[204]
CU-76 CU-32	Inhibit the dimerization of cGAS	Specifically target the inhibition of the cGAS–STING pathway	[205]
Aspirin	Acetylates Lys384, Lys394, and Lys414 amino acid residues of cGAS	Suppresses immune responses in AGS mouse models	[71]
Compound C	Reduces cGAMP accumulation	Suppresses type-I IFN induction	[206]
DNA-binding inhibitors			
AMDs	Block cGAS–dsDNA interaction	The IC_50_ in THP-1 cells is 3–25 μM	[207,208]
Suramin	Disrupts the formation of cGAS–dsDNA complex	Modulates type-I IFN level in THP-1 cells	[209]
A151	Competitively binds to DNA-binding domain of cGAS	Inhibits type-I IFN signaling in *TREX1*-deficient cells	[210]
STING inhibitors			
CDN-binding site inhibitors			
Astin C	Inhibits the recruitment of IRF3 to the STING-TBK1 signalosome	Inhibits the expression of *Ifnb*, *Cxcl0*, and *Tnf* mRNA in multiple tissues of mouse models	[211]
THIQ	Transforms STING into an inactive, open conformation	Inhibits cGAMP-induced type-I IFN secretion in THP-1 cells	[212]
Palmitoylation inhibitors			
Nitrofurans	Bind to Cys91 to inhibit the palmitoylation of STING	Strongly suppress inflammatory response in mouse models	[213]
Indole urease	Forms a covalent bond with Cys91 of STING	Reduces the production of type-I IFN in *TREX1*-deficient mouse tumor model	[213]
NO_2_-FAs	Covalently modify STING in Cys88 and Cys91	Reduce type-I IFN production in response to HSV-2 infection in THP-1 cells	[214]

**Abbreviations:** BMDMs, bone marrow-derived macrophages; IC_50_, half-maximal inhibitory concentration; primary H-macrophages, primary human macrophages; AMDs, antimalarial drugs (such as hydroxychloroquine and quinacrine); THP-1 cells, human myeloid leukemia mononuclear cells; THIQ, tetrahydroisoquinoline; NO_2_-FAs, nitro fatty acids; cGAS, cyclic GMP-AMP synthase; STING, stimulator of interferon genes; type-I IFN, type-I interferon; AGS, Aicardi–Goutières syndrome; cGAMP, cGMP-AMP; dsDNA, double-stranded DNA; CDNs, cyclic dinucleotides; IRF3, interferon regulatory factor-3; TBK1, TANK-binding kinase 1.

**Table 6 ijms-24-13316-t006:** Agonists of STING.

Compound	Mechanisms	Preclinical Effects	Refs
CDN agonists			
Natural CDNs	Activate APCs and CD8+ T cells	Enhance antitumor signals	[215]
ADU-S100	Activates all STING variants and improves their stability	Induces durable tumor regression	[215]
cGAMP-NPs	Inserted liposomal NPs can deliver cGAMP more efficiently	Create anti-tumor microenvironment	[216]
Non-CDN agonists			
DMXAA	Higher affinity than cGAMP and activates STING efficiently	Restricts tumorigenesis	[217]
Zebularine	Enhances the STING gene expression by demethylation	Reduces tumor burden	[218]
G10	Stabilizes the structure of STING	Suppresses tumor growth	[219]
ABZI	Induces type-I IFN production that is 400 times higher than that of cGAMP	Tumor volume regression	[220]
DSDP	Induces STING-dependent cytokine responses	Triggers antiviral responses	[221]
BNBC	Specifically activates STING	Triggers antiviral responses	[222]
Indirect agonists			
Radiotherapy	Causes DSBs and the accumulation of cytosolic DNA	Activates adaptive immune response	[223]
Cisplatin	Inhibits DDR and the release of chromatin fragments into the cytosol	Activates CD8^+^ T cells	[43]
Teniposide	Induces DNA damage in cancer cells	Activates DCs and T cells	[224]

**Abbreviations:** CDNs, cyclic dinucleotides; APCs, antigen-presenting cells; cGAMP-NPs, cGAMP-nanoparticles; DMXAA, 5, 6-dimethylxanthenone-4-acetic acid; ABZI, amide benzimidazole; DSDP, dispiro diketopiperzine; BNBC, 6-bromo-N-(naphthalen-1-yl)benzo[d][1,3] dioxole-5-carboxamide; STING, stimulator of interferon genes; cGAMP, cGMP-AMP; type-I IFN, type-I interferon; DSBs, double-strand breaks; DCs, dendritic cells.

## Data Availability

No new data were created or analyzed in this study. Data sharing is not applicable to this article.

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
