# Peer review of "Significance of the cGAS-STING Pathway in Health and Disease"

_ijms, 2023, doi:10.3390/ijms241713316_

Round 1

Reviewer 1 Report

This is a comprensive review on the critical role of the cGAS/STING pathway in the regulation of immunity leading to the development of immune-related and inflammatory diseases. The authors provide a brief  summary of the current state of clinical and nonclinical development of modulators targeting cGAS/STING as new therapeutic strategies for infectious, immune and inflammatory diseases. The paper is well balanced and addresses the most important issues regarding the subject.

Author Response

We are so grateful for the professional review work and positive attitudes to our article, we will endeavor to produce higher quality articles and then submit in the journals belongs to MDPI given the opportunity.

Reviewer 2 Report

Comments to the Authors:

The manuscript by Zhou et al. is well-written and provides a comprehensive understanding of the cGAS-STING pathway in health and disease. I have a few suggestions to consider.

Tables 1 and 2 could be organized based on the target of the pathway rather than modification. For example, group the information for cGAS together, providing details on its modifications and the modulators that target cGAS, and similarly for STING and other components. This may improve the clarity of the presentation.

Some crucial information is missing in Table 1, such as the regulation of STING by ULK1 (PMID: 24119841) and STING degradation by RNF5 (PMID: 19285439). Please review the information and update the table to provide a more comprehensive view of pathway regulation.

In Table 2, consider including the information about cGAS monoubiquitination by TRIM41 (PMID: 29760876) and the protection of cGAS by TRIM14 (PMID: 27666593).

Figure 2 contains an overload of concepts, making it confusing to understand. It would be helpful to create clear demarcations among cell death, autophagy, and senescence within the figure to enhance clarity.

Overall, the manuscript presents valuable insights into the cGAS-STING pathway, and these suggestions aim to improve the visual presentation and completeness of the information.

Author Response

At first, it is necessary to express my appreciations to your valuable opinions and comments, it helps me acquire latest knowledge regarding to the cGAS-STING pathway as well as provides me additional insights into the modulation pattern of cGAS-STING pathway. In the next, please allow me to explain my revisions accepting your suggestions at below:

Suggestion 1: Tables 1 and 2 could be organized based on the target of the pathway rather than modification. For example, group the information for cGAS together, providing details on its modifications and the modulators that target cGAS, and similarly for STING and other components. This may improve the clarity of the presentation.

Explanation 1: I have been grouped the information for modifications and modulators targeting cGAS, cGAMP and STING together respectively, which were marked with blue color in the Table 1 and Table 2. Please check, Thank you.

Suggestion 2: Some crucial information is missing in Table 1, such as the regulation of STING by ULK1 (PMID: 24119841) and STING degradation by RNF5 (PMID: 19285439). Please review the information and update the table to provide a more comprehensive view of pathway regulation.

Explanation 2: I have been added the modulation patterns and further information of that ULK1 and RNF5 regulates STING in the innate immunity in Table 1 marked with blue color. Please check, Thank you.

Suggestion 3: In Table 2, consider including the information about cGAS monoubiquitination by TRIM41 (PMID: 29760876) and the protection of cGAS by TRIM14 (PMID: 27666593).

Explanation 3: I have been included the detailed information regarding to how TRIM41 modulates cGAS in Table 2 marked with blue color. Please check, Thank you.

Suggestion 4: Figure 2 contains an overload of concepts, making it confusing to understand. It would be helpful to create clear demarcations among cell death, autophagy, and senescence within the figure to enhance clarity.

Explanation 4: I have been made clear demarcation through dividing cell death, autophagy and senescence into 3 parts using dotted lines. Please checked, Thank you.

Reviewer 3 Report

This is a timely and well written review on cGAS/STING and its importance in health and disease. Overall, the review is well described in a logical order and provides a good overview of the field, including the comprehensive tables summarizing the major findings. However, there are several points that need to be addressed before publication:

Line 73ff: cGAS is not only a cytoplasmic, but a nuclear sensor as well. For instance, Volkman et al 2019 show that cGAS is predominantly a nuclear protein. All the recent discoveries on nuclear cGAS, its restriction by BAF and tethering to nucleosomes should be described to cover the whole breath of information, particularly, as these points are very important for the role of cGAS in health and disease, see Liu et al (Ref 172: nuclear cGAS promotes tumorigenesis). Literature such as Guey et al (2020) and Boyer et al., Kuljirai et al., Michalski et al., Pathare et al, Zhao et al on the structure of cGAS tethered to nucleosomes should be at least mentioned…

The authors should check the literature throughout the manuscript for proper assignment: the first and major papers describing the respective results should be cited! as a reviewer, I cannot spend the time to check every single literature citation, the examples below and above were the ones that immediately caught my attention, therefore, the authors should check whether the key literature throughout the manuscript is present.  

Just as examples : for instance for the statement line 111, reference 13 is cited, however, the first and major publications on this are by Zhang et al., Cell Reports 2014 and Li et al., immunity 2013: another example: ladder-like structure: here, definitely Andreeva et al., Nature 2017 is missing!, Liquid phase transition besides Du et al., please cite also Xie et al 2019..

Paragraph 4:

The authors should include new literature insights on HIV recognition by cGAS, co-sensor or adaptors of cGAS that are required for sensing of pathogens (Yoh et al., Mol Cell 2022; Yoh et al., Cell 2015)

The authors should mention the AP-1 control of the termination of STING activation and palmitoylation to finalize the list of regulators of STING pathway.  

Paragraph 8:

In order to complete this paragraph, particularly as the authors already mention the RNA viruses that possess strategies to block cGAS/STING signaling, the authors should mention that RNA viral infections (and NOT only DNA viruses such as HSV) can induce mtDNA leakage and therefore sensing through cGAS /STING as shown with e.g. Dengue Virus (Aguierre et al, 2017), SARS-CoV-2 and others..

Line 388ff: this sentence is outright wrong: “In the case of retro-viruses (including the typical HIV-1 retrovirus), cGAS cannot theoretically sense these viruses because their genome does not contain dsDNA sequences….. “ first of all, the authors should mention all nucleic acids that theoretically can be bound by cGAS (e.g. long DNA-RNA hybrids), secondly, in case of retroviruses, it has been shown that RT-products induce cGAS-dependent innate immune responses (which could include anything from dsDNA up to DNA/RNA hybrids; see e.g. review Yin et al., Cells 2019), Gao et al, 2013 and Li et al., 2013 first reported sensing of HIV-1; RT products are recognized by PQBP1 and cGAS (Yoh et al 2015, 2022). The nature of the RT product is not yet defined.

Paragraph 9:

AGS: to date 9 genes (and not 7!!) have been attributed to AGS, including mutations in LSM11 and RNU7-1 (and here: nuclear activation of cGAS!!): see Uggenti et al., 2020

New results should be mentioned: cGAS-Sting driving ageing-related inflammation and neurodegeneration (Gulen et al., 2023)

Author Response

I sincerely thank you for your valuable, careful and professional feedback which helps us improve the quality of our manuscript. Your comments bring me concrete benefit in how to write a precise, comprehensive academic article regarding to the cGAS-STING pathway. Thank you very very much. Subsequently, please let me demonstrate my answers relating to questions which needs to be addressed.

Question 1: Line 73ff: cGAS is not only a cytoplasmic, but a nuclear sensor as well. For instance, Volkman et al 2019 show that cGAS is predominantly a nuclear protein. All the recent discoveries on nuclear cGAS, its restriction by BAF and tethering to nucleosomes should be described to cover the whole breath of information, particularly, as these points are very important for the role of cGAS in health and disease, see Liu et al (Ref 172: nuclear cGAS promotes tumorigenesis). Literature such as Guey et al (2020) and Boyer et al., Kuljirai et al., Michalski et al., Pathare et al, Zhao et al on the structure of cGAS tethered to nucleosomes should be at least mentioned…

Answer 1: I have been described the works of Volkman et al. 2019, Liu et al., Guey et al., Boyer et al., Kuljirai et al., Michalski et al., Pathare et al, and Zhao et al in line 73ff or near line 73ff with red color, which provides detailed information about the intracellular location, function and role of cGAS in health and disease Please check, Thank you.

Question 2: The authors should check the literature throughout the manuscript for proper assignment: the first and major papers describing the respective results should be cited! As a reviewer, I cannot spend the time to check every single literature citation, the examples below and above were the ones that immediately caught my attention, therefore, the authors should check whether the key literature throughout the manuscript is present.  

Just as examples: for instance for the statement line 111, reference 13 is cited, however, the first and major publications on this are by Zhang et al., Cell Reports 2014 and Li et al., immunity 2013: another example: ladder-like structure: here, definitely Andreeva et al., Nature 2017 is missing!, Liquid phase transition besides Du et al., please cite also Xie et al 2019.

Answer 2: I spent a lot of time checking the accuracy of references citation, I can ensure that key references throughout the manuscript is present. In the meanwhile, there is no missing citation of literatures. At all, the missing references in the examples were cited. the new cited part were marked with red color. Please check, Thank you.

Question 3:

Paragraph 4:

The authors should include new literature insights on HIV recognition by cGAS, co-sensor or adaptors of cGAS that are required for sensing of pathogens (Yoh et al., Mol Cell 2022; Yoh et al., Cell 2015)

The authors should mention the AP-1 control of the termination of STING activation and palmitoylation to finalize the list of regulators of STING pathway.  

Answer 3: According to these references (Yoh et al., Mol Cell 2022; Yoh et al., Cell 2015) and the references which I have mentioned in the article, I have been drew a comprehensive mechanism of how cGAS and its co-sensor or adaptors activates downstream signaling cascade and innate immunity to against pathogens infection, such as HIV with red color in paragraph 4. At the sametime, I have mentioned the detailed information about how AP-1 regulates STING in the Table 2 and 2-BP suppresses the palmitoylation of STING with red color. Please check, Thank you.

Question 4:

Paragraph 8:

In order to complete this paragraph, particularly as the authors already mention the RNA viruses that possess strategies to block cGAS/STING signaling, the authors should mention that RNA viral infections (and NOT only DNA viruses such as HSV) can induce mtDNA leakage and therefore sensing through cGAS /STING as shown with e.g. Dengue Virus (Aguierre et al, 2017), SARS-CoV-2 and others..

Line 388ff: this sentence is outright wrong: “In the case of retro-viruses (including the typical HIV-1 retrovirus), cGAS cannot theoretically sense these viruses because their genome does not contain dsDNA sequences….. “ first of all, the authors should mention all nucleic acids that theoretically can be bound by cGAS (e.g. long DNA-RNA hybrids), secondly, in case of retroviruses, it has been shown that RT-products induce cGAS-dependent innate immune responses (which could include anything from dsDNA up to DNA/RNA hybrids; see e.g. review Yin et al., Cells 2019), Gao et al, 2013 and Li et al., 2013 first reported sensing of HIV-1; RT products are recognized by PQBP1 and cGAS (Yoh et al 2015, 2022). The nature of the RT product is not yet defined.

Answer 4:

I have been listed different types of virus which can block cGAS-STING pathway in table 3. At the sametime, I mentioned the detailed mechanisms of mtDNA leakage-caused cGAS-STING pathway under other DNA or RNA virus infection, such as dengue virus, SARS-CoV-2 and so forth with red color in paragraph 8. Please check, thank you.

I have been revised the line 388ff wrong sentence to the correct version through the references you mentioned with red color. Please check, thank you.

Question 5:

Paragraph 9:

AGS: to date 9 genes (and not 7!!) have been attributed to AGS, including mutations in LSM11 and RNU7-1 (and here: nuclear activation of cGAS!!): see Uggenti et al., 2020

New results should be mentioned: cGAS-Sting driving ageing-related inflammation and neurodegeneration (Gulen et al., 2023)

Answer 5:

I have been updated that 9 genes have been attributed to AGS, including mutations in LSM11 and RNU7-1 with red color in paragraph 9. At the meanwhile, I mentioned the latest results according to article “cGAS-Sting driving ageing-related inflammation and neurodegeneration” in detail with red color and illustrated the mechanism of it in Figure 3. Please check, thank you.

Reviewer 4 Report

Congratulations! A very interesting review summarizing all recent scientific data regarding cGAS-STING pathway. It worths being published in the journal. I read it with great interest. Also, i liked figures. 

Author Response

Responses to reviewer 4(Round 1):

I am very appreciating to your time involved in reviewing the manuscript and your very encouraging and positive comments on the merits. No matter what, I will devote myself to do research on cGAS-STING pathway, unveiling the mysterious role of cGAS-STING pathway in innate immunity. In addition, we will do our best to produce higher-quality reviews or research articles publishing in the journal of MDPI given the opportunity. 

Round 2

Reviewer 3 Report

The authors made a significant effort to incorporate all my comments and suggestions. Most of the changes made in the revised manuscript are adequate, except one paragraph listed below. This paragraph need to be corrected, as it is unfortunately not entirely correct

 line 400-406:

In the case of retroviruses (including the typical HIV-1 retrovirus), cGAS is recruited to the capsid of HIV-1 through binding of polyglutamine-binding protein 1 (PQBP1) which is the adaptor of HIV-1, and then induces STING-dependent anti-viral immune response by recognizing the unique Y-DNA in the stem-loop structure of RNA-DNA HIV-1 hybrids which is synthesized in  early reverse transcripts [20, 100,217,218].”

Please change “…adaptor of HIV-1..” to “.. adaptor of cGAS”  

Please delete the last part of this paragraph, please delete “by recognizing the unique Y-DNA in the stem-loop structure of RNA-DNA HIV-1 hybrids which is synthesized in  early reverse transcripts”, this statement is a misinterpretation of the previous described statement of PQBP1-cGAS recruitment, it has not been shown and is as yet unknown

Author Response

Responses to reviewer 3 (Round 2)

Please allow me to express my appreciations again to your careful and professional feedbacks of our manuscript. I am so sorry about my cursoriness in the paragraph which you commented. I have been revised these mistakes according to your suggestions, and then please let me explain my revisions accepting your suggestions at below:

Suggestion 1:

 line 400-406:

“In the case of retroviruses (including the typical HIV-1 retrovirus), cGAS is recruited to the capsid of HIV-1 through binding of polyglutamine-binding protein 1 (PQBP1) which is the adaptor of HIV-1, and then induces STING-dependent anti-viral immune response by recognizing the unique Y-DNA in the stem-loop structure of RNA-DNA HIV-1 hybrids which is synthesized in  early reverse transcripts [20, 100,217,218].”

Please change “…adaptor of HIV-1..” to “.. adaptor of cGAS”  

Answer 1:

I have been changed “adaptor of HIV-1” to “adaptor of cGAS” with purple color in line 400-406, please check, thank you.

Suggestion 2:

Please delete the last part of this paragraph, please delete “by recognizing the unique Y-DNA in the stem-loop structure of RNA-DNA HIV-1 hybrids which is synthesized in early reverse transcripts”, this statement is a misinterpretation of the previous described statement of PQBP1-cGAS recruitment, it has not been shown and is as yet unknown

Answer 2: I have been deleted “by recognizing the unique Y-DNA in the stem-loop structure of RNA-DNA HIV-1 hybrids which is synthesized in early reverse transcripts”. Please check, thank you.